# *i*-Propylammonium Lead Chloride Based Perovskite Photocatalysts for Depolymerization of Lignin Under UV Light

**DOI:** 10.3390/molecules25153520

**Published:** 2020-07-31

**Authors:** Samia Kausar, Ataf Ali Altaf, Muhammad Hamayun, Nasir Rasool, Mahwish Hadait, Arusa Akhtar, Shabbir Muhammad, Amin Badshah, Syed Adnan Ali Shah, Zainul Amiruddin Zakaria

**Affiliations:** 1Department of Chemistry, University of Gujrat, Hafiz Hayat Campus, Gujrat 50700, Pakistan; samiakosar93@gmail.com (S.K.); hamayunf@uog.edu.pk (M.H.); atafali83@gmail.com (M.H.); arusachaudry@gmail.com (A.A.); 2Department of Chemistry, University of Okara, Okara 56300, Pakistan; 3Department of Chemistry, Government College University, Faisalabad 38000, Pakistan; nasirrasool@gcuf.edu.pk; 4Research Center for Advanced Material Science (RCAMS), King Khalid University, P.O. Box 9004, Abha 61413, Saudi Arabia; shabbir193rb@gmail.com; 5Department of Physics, College of Science, King Khalid University, P.O. Box 9004, Abha 61413, Saudi Arabia; 6Department of Chemistry, Quaid-i-Azam University, Islamabad 45320, Pakistan; aminbadshah@qau.edu.pk; 7Faculty of Pharmacy, Universiti Teknologi MARA Cawangan Selangor Kampus Puncak Alam, Bandar Puncak Alam 42300, Selangor D. E., Malaysia; syedadnan@uitm.edu.my; 8Atta-ur-Rahman Institute for Natural Products Discovery (AuRIns), Universiti Teknologi MARA Cawangan Selangor Kampus Puncak Alam, Bandar Puncak Alam 42300, Selangor D. E., Malaysia; 9Department of Biomedical Science, Faculty of Medicine and Health Sciences, Universiti Putra Malaysia, Serdang 43400, Selangor, Malaysia; 10Halal Institute Research Institute, Universiti Putra Malaysia, Serdang 43400, Selangor, Malaysia

**Keywords:** photocatalysis, lignin depolymerization, perovskites, kinetics, activation energy

## Abstract

Lignin depolymerization for the purpose of synthesizing aromatic molecules is a growing focus of research to find alternative energy sources. In current studies, the photocatalytic depolymerization of lignin has been investigated by two new iso-propylamine-based lead chloride perovskite nanomaterials (**SK9** and **SK10**), synthesized by the facile hydrothermal method. Characterization was done by Powder X-Ray Diffraction (PXRD), Scanning Electron Microscopy (SEM), UV-Visible (UV-Vis), Photoluminescence (PL), and Fourier-Transform Infrared (FTIR) Spectroscopy and was used for the photocatalytic depolymerization of lignin under UV light. Lignin depolymerization was monitored by taking absorption spectra and catalytic paths studied by applying kinetic models. The %depolymerization was calculated for factors such as catalyst dose variation, initial concentration of lignin, and varying temperatures. Pseudo-second order was the best suited kinetic model, exhibiting a mechanism for lignin depolymerization that was chemically rate controlled. The activation energy (E_a_) for the depolymerization reaction was found to be 15 kJ/mol, which is remarkably less than conventional depolymerization of the lignin, i.e., 59.75 kJ/mol, exhibiting significant catalytic efficiencies of synthesized perovskites. Products of lignin depolymerization obtained after photocatalytic activity at room temperature (20 °C) and at 90 °C were characterized by GC-MS analysis, indicating an increase in catalytic lignin depolymerization structural subunits into small monomeric functionalities at higher temperatures. Specifically, 2-methoxy-4-methylphenol (39%), benzene (17%), phenol (10%) and catechol (7%) were detected by GC-MS analysis of lignin depolymerization products.

## 1. Introduction

Society’s dependence on non-renewable energy sources and increasing demand for fuels and chemicals are the most addressed concerns of researchers for the purpose of developing sustainable technologies that would probably enable the efficient consumption of renewable resources [1,2,3,4]. Such a striking source material is lignocellulose, which is produced in substantial quantities from agricultural and forestry activity worldwide [5]. Lignin is the major renewable energy source in nature for aromatic building blocks and has the utmost potential to produce bulk/functionalized aromatic compounds and to offer suitable alternatives to universally employed petroleum-derived BTX (benzene, toluene, and xylene) [6,7,8]. 

Lignin conversion is a potential challenge because of its high molecular weight and polymeric nature [9,10,11]. The quest for novel catalytic methods to transform polymeric substrates like lignin into value-added compounds initiated tremendous research in various fields, such as homogeneous catalysis [12,13,14] and heterogeneous catalysis [15]. Several chemical and biological pathways like thermochemical [16], oxidative [9], or biochemical [17] depolymerization methods have been investigated for the conversion of lignin to various product classes, but these are not extensively used, owing to their high operating costs [18]. 

Over the past ten years, light-mediated photochemistry has observed tremendous developments [19,20,21]. The utilization of photon energy for chemical transformations offered versatile chemical transformations, empowering numerous reactions and compounds that were previously unreachable through traditional methods over the past decade [22,23,24]. Hence, photocatalysis [25] is one of the advanced techniques that has applications in various fields of green energy, medicine, chemical synthesis, and environmental technology, etc., [26]. Therefore, photocatalytic depolymerization of lignin is likely to be the most environment friendly, operative, and capable method because of its productivity, energy-saving, and low cost [27].

A lot of research interest is growing in organic/inorganic hybrid alkyl lead halide perovskites (RPbX3), due to their most promising catalytic, energy, and light-harvesting applications [28]. They have been widely investigated for a wide range of light-harvesting applicabilities, e.g., photovoltaics [29], light-emitting diodes (LED) or lasers [30] and radiation detection [31]. MAPbX3 solar cells have had excellent developments in efficiency in comparison to any other solar cells since their invention [32]. Metal halide-based perovskites are cost-effective materials processable in the solution, having outstanding intrinsic properties that make them appropriate candidates for light harvesting technologies in the future [33]. In particular, Pb-based hybrid perovskites have emerged as potential competitors in photovoltaics and light harvesting applications due to their high absorption coefficient, direct bandgap, long carrier lifetime, and enhanced balanced hole and electron mobility characteristics [34,35,36]. 

Various photoactive transition-metal oxides, metal oxide-based perovskites, and organic chromophores were developed to catalyze the transformations of high molecular weight substrates like lignin and have been reported previously [37,38,39]. However, reports on crystalline RPbX3 for the photocatalytic transformation of high molecular weight polymeric substrates are relatively few. Metal oxide-based perovskites are being employed for lignin depolymerization, but alkyl amine-based perovskites are rarely reported for biomass conversion. Considering the importance of the subject, this study was intended to synthesize the lead chloride perovskite materials **SK9** and **SK10** with new alkyl moiety (iso-propylamine) for photocatalytic depolymerization of lignin not studied before. 

## 2. Results and Discussion

### 2.1. Characterization 

#### 2.1.1. Powder X-ray Diffraction (PXRD) and Scanning Electron Microscopic (SEM) Analysis 

The phase composition and particle size of synthesized perovskite materials (**SK9** and **SK10**) were analyzed by PXRD and the data evaluated are listed in Table 1. Figure 1 shows Powder XRD patterns of synthesized IAPbCl_3_ perovskite particulates milled from a piece of corresponding large-sized crystal. The powder XRD of the crystalline IAPbCl_3_ perovskite material **SK9** shows diffraction peaks at 22.01, 35.22, 39.34, 41.67, 51.04 2𝜃 values, whereas **SK10** shows at 23.2, 39.4, 41.63, 51.1 2𝜃 values. Observed intensities at 22–23, 35.22, 39–41 ranges of 2𝜃 values correspond to the (100), (110), (200) lattice planes of cubic structure, most probably at room temperature [40,41,42]. Powder XRD analysis indicated that the crystal structure of the IAPbCl_3_ perovskites (**SK9** and **SK10**) with the cubic system (space group Pm3m, a = b = c = 5.6855 Å) at room temperature had quite similar intensities at corresponding 20 values to that reported in the literature, as given in Figure 1a–c [36]. 

Other parameters from PXRD, i.e., crystallite size, dislocation densities, crystallite volume, and microstrain, were evaluated for **SK9** and **SK10**, given in Table 1. Particle size calculation was performed by employing the Debye–Scherrer Equation (1) [43].
D = 0.91λ/βcos 𝜃(1)
where D is the crystallite size of the material and λ is the wavelength of the X-ray beam with 1.54 Å value. The cos θ is an angle at which diffraction occurs and β represents the value at FWHM (full width at half maximum) [44].

The average crystallite size calculated at their corresponding intensities was found to be 77.10 nm for **SK9** and 75.59 nm for **SK10**. The dislocation density (δ) is defined as the length of dislocation lines per unit volume of the crystal, which explains the measure of defects in the material and was calculated using the Equation (2) [45].
δ = 1/D^2^(2)

The volume of crystalline (V) is measured, employing the relation (**V = D^3^**) [46], whereas the strain-induced broadening in materials because of imperfection in crystals and distortion in them is calculated as microstrain (ε) [47]. Positive values of ε indicate the tensile strain, whereas a negative value gives a compressive type of strain. Microstrain in synthesized perovskites was calculated following Equation (3) [47]:ε = β/4tan 𝜃(3)

Very low positive values of microstrain and dislocation densities reveal the extent of defect found to be less in perovskite materials (**SK9** and **SK10**), showing greater stability and negligible distortion or crystal imperfections. 

SEM parameters calculated for perovskite nanomaterials (**SK9 and SK10**) are given in Table 1, whereas SEM images of **SK9** and **SK10** are depicted in Figure 2. Images indicated homogenous morphology incorporating very broad elliptical leaflet like structures for both materials, having somewhat rugged surface appearance. The broad elliptical structure of perovskites can play an important role in providing a greater surface for the adsorption of high molecular weight substrate for photocatalytic depolymerization. 

#### 2.1.2. UV and Photoluminescence (PL) Analysis 

UV-Visible spectra of perovskite nanomaterials (**SK9**–**SK10**) and PbCl_2_ are shown in (Figure 3). **SK9** and **SK10** prepared with different concentrations of iso-propyl amine (4 mL in **SK9** and 5 mL in **SK10**) have shown higher absorption than PbCl_2_ at 209 nm wavelength. Extended absorption in the UV range, i.e., down to 280 nm, signifies the interesting optical characteristics of these materials [48]. It is seen from the figure that both perovskite materials show higher absorption intensities at 209 nm as compared to PbCl_2_, whereas increasing the concentration of IA in **SK10** (5 mL), there is a significant decrease in absorption peak intensity in the UV light range. High absorption intensity corresponds to using a small amount of IA (4 mL) in **SK9**, indicating the capability of the material to be employed in catalytic and optoelectronic applications. High absorption intensities of perovskite catalysts may play an important role for photocatalytic activities as the subject of this study is to depolymerize lignin using UV as light source. 

Tauc plots for the direct bandgap have been drawn with the help of the absorption spectra of **SK9** and **SK10**, given in Figure 4. The bandgap values were obtained through Tauc plots [49], as in Equation (4).
(αhυ)^2^ = A (hυ − E_g_)(4)
where h is called Planck’s constant, ν is frequency, E_g_ is optical bandgap, A is the energy-independent constant. The exponent ‘2′ explains the direct transitions, in this case, as this exponent has different values for different kinds of transitions [50]. The calculated band gap energy for **SK9** and **SK10** is 5.6 and 5.3 eV, respectively, which indicate that the samples can absorb light in UV to the visible region and can be utilized as efficient photocatalytic applications [50,51,52]. 

PL emission spectra of **SK9** and **SK10** have been obtained through a PL spectrophotometer with an exciton wavelength of 450 nm. Figure 5 shows the PL spectra of the PbCl_2_ giving a sharp peak at 532 nm, whereas synthesized perovskites **SK9** and **SK10** exhibiting prominent PL peaks at 469, 488, 532, and 677 nm correspond to 2.6, 2.5, 2.3 and 1.83 eV energy range. An increase in the emitted intensity in perovskites **SK9** and **SK10** in comparison to lead chloride was observed. The narrower PL peaks also designate a lower trap density in the synthesized materials, which is beneficial for them to be applied in solar cells and as efficient catalysts [53]. High emission intensities may correspond to significant photocatalytic efficiencies, as higher emissions specify low trap densities. Hence, synthesized perovskites may show excellent catalytic behaviors.

#### 2.1.3. FTIR Analysis 

The Fourier-Transform Infrared (FTIR) analysis of synthesized perovskites **SK9** and **SK10** along with PbCl_2_ is given in Figure 6 (some characteristic peaks are highlighted with marks in the figure). Generally, perovskite nanomaterials show absorption bands in the Raman and fingerprint region of IR due to interatomic vibrations which lie below 1000 cm^−1^, hence, most of the peaks in IR spectra are due to organic (iso-propylamine) moiety. Spectra showed almost similar peaks with reported methyl amine-based lead chloride perovskites [54]. The strong absorption band at the wavenumber about 3400 cm^−1^ is due to the presence of OH- in PbCl_2_ [55]. The small peaks at 974–980 cm^−1^ were originated from C-N stretching in **SK9** and **SK10**. The intense peak near 1000 cm^−1^ was associated with the CH_3_^+^ rocking vibration. 

### 2.2. Photocatalytic Depolymerization of Lignin Under UV Light 

For all sample aliquots taken during the photocatalytic lignin (**L1**) depolymerization activity at different time intervals, absorbance at 280 nm was measured by UV spectra. The %lignin depolymerization was evaluated from the absorbance, as explained in Section 3.3. Abs_280_ corresponds to the aromatic rings of lignin structure, and the decrease in it signifies the alteration of the aromatic ring structure [56].

With the increasing photocatalytic activity time duration, the absorption peaks at 280 nm tend to decrease, representing the transformation of the aromatic backbone of lignin. Likewise, various factors, i.e., effect of catalytic dosage, starting lignin concentrations, and varying temperature, were studied at various time intervals by calculating %lignin depolymerization. Perovskites exhibited high photocatalytic efficiency with an increase in %lignin depolymerization, with increasing time intervals for all the factors studied. 

#### 2.2.1. Effect of Time

Lignin depolymerization was studied for **SK9** and **SK10** at different time intervals, i.e., 10, 20, 30, 40, 50, and 60 min, as shown in Figure 7 and %depolymerization was calculated, given in Table 2. **SK9** and **SK10** exhibited proficient catalytic efficiency to depolymerize the 0.1 g/L lignin. After every 10 min interval of irradiation under UV light, a decrease in absorption intensity of the lignin solution was observed, as shown in Figure 7. A total of 17% depolymerization was calculated after 10 min of UV light irradiation, which increased to 31, 48, 72, 79, 86.5% after 20, 30, 40, 50 and 60 min, respectively, for **SK9**. Meanwhile, 10% depolymerization was calculated after 10 min of UV light exposure, which increased to 20, 31, 45, 65.5, 72.5% after 20, 30, 40, 50, and 60 min, respectively, for **SK10**. The proficient catalytic efficiencies of **SK9** and **SK10** perovskites to depolymerize lignin were credited to their larger surface areas, which provided a greater number of active sites. Correlation between the experimental data of lignin depolymerization was calculated to be very significant, as R^2^ values were close to 1 (Figure 8). **SK9** was found more catalytically efficient than **SK10**, hence, other factors were studied with **SK9**.

#### 2.2.2. Effect of Catalyst Dosage

Effect of the catalyst **SK9** dosage on the lignin depolymerization was analyzed by variating catalyst dose from 0.025 to 0.1 g/L for 60 min irradiation under UV light. After every 10 min interval of irradiation under UV light, a decrease in absorption intensity of the lignin solution was observed at each catalyst dosage, as shown in Figure 9. The %depolymerization of lignin (100 ppm of **L1**) was calculated for different doses of **SK9** at various time intervals, as given in Table 3. A decrease in absorption intensity, hence, increase in %depolymerization, was found with increasing time intervals from 10 to 60 min for each catalyst dose, i.e., 0.025, 0.05, 0.075 and 0.1 g/L of catalyst dose, as shown in Appendix A. 

In total, 41.5% lignin depolymerization was calculated for 0.025 g **SK9**, 67% for 0.05 g, and 73% by using 0.075 g **SK9** after 60 min of UV light irradiation to lignin solution. The maximum lignin depolymerization (86.5%) was attained at 60 min of UV light exposure for dosage 0.1 g/L of **SK9**. A total of 17% depolymerization was calculated after 10 min of UV irradiation, which increased up to 31, 48, 72, 79, and 86.5% after 20, 30, 40, 50, and 60 min respectively for 0.1 g **SK9**. The maximal increase in the dose of **SK9** (0.1 g/L) resulted in maximum lignin depolymerization, most possibly because of more surface given by the catalyst, which caused a greater number of photons to interact at the catalyst’s surface and increase the passage of UV irradiation through the **L1** solution. The correlation between the %lignin depolymerization and the catalyst dosage was calculated to be significant as R^2^ values were close to 1 (Figure 10). 

#### 2.2.3. Effect of Initial Lignin Concentration

The effect of starting concentrations of the lignin on its depolymerization was studied for 0.1 g **SK9**, with the lignin concentration variating from 50 to 200 ppm and investigating the %lignin depolymerization. After every 10 min interval of irradiation under UV light, a decrease in absorption intensity of the lignin solution was observed at each lignin concentration, as shown in Figure 11 (detailed in Appendix A). Photocatalytic %lignin depolymerization by **SK9** was observed to decrease with the increase in lignin concentration from 50 to 200 ppm, listed in Table 4. In total, 92.5% of lignin depolymerization was achieved after 60 min of irradiation of UV light at a 50 ppm initial lignin concentration, which was decreased as 86.5% for 100 ppm, 73% for 150 ppm and 62.5% for 200 ppm. 

The lower efficiency observed going towards high lignin concentration is possibly ascribed to the greater number of lignin molecules adsorbed on the catalyst’s surface. Adsorption of the number of molecules caused a significant decrease in the availability of active sites on the catalyst’s surface, hence, a smaller number of active species generated. Besides this, increasing lignin concentration resulted in a significant decrease in the number of photons reaching the catalyst’s surface. This caused less UV light availability to excite the particles of the catalyst due to significant absorption by the lignin molecules. The correlation constant R^2^ between the %depolymerization of the lignin concentrations by **SK9** at the different time intervals was calculated to be close to 1, as given in Figure 12.

#### 2.2.4. Effect of Variating Temperature

The dependence of rate constant of lignin depolymerization on temperature was investigated by 0.1 g **SK9** (using 100 ppm of **L1**) at 20, 35, and 65 and 90 °C. After every 10 min interval of irradiation under UV light, a decrease in absorption intensity of the lignin solution was observed at each increase in temperature, as depicted in Figure 13 (detailed in Appendix A). %Lignin depolymerization was evaluated at different time intervals at each temperature. Depolymerization reaction rate along with %lignin depolymerization tended to increase with the increasing range of temperature, as given in Table 5. Maximum %depolymerization was obtained at 90 °C with 0.1 g **SK9**, which showed an increase in %depolymerization with the increasing temperature. In total, 86.5% lignin depolymerization was calculated at room temperature (20 °C), whereas 89% for 35 °C, 92% for 65 °C and 97.5% for 90 °C temperature was achieved. The increase in %depolymerization of lignin upon increasing temperature range was mainly because of the thermal breakdown of the aromatic backbone of the lignin crosslinked structure. Going to high temperatures caused the breakdown of crosslinked functionalities, resulting in depolymerization into monomeric compounds [57]. Correlation between %lignin depolymerization by **SK9** at various temperatures was calculated to be close 1, as given in Figure 14.

### 2.3. Kinetic Analysis 

To determine adsorption kinetics of liquid/solid systems, pseudo kinetic models i.e., Pseudo-first order and pseudo-second order kinetics are often employed [58,59,60]. Kinetics of depolymerization of the lignin on the solid perovskite’s surface was demonstrated best by the pseudo-second order kinetic model by following Equation (5):t/C_t_ = 1/k_2_C_t_^2^ + t/C_o_(5)
where C_t_ is the concentration of the lignin solution at a specific time interval, C_o_ is the initial lignin concentration, t is the time at which sample aliquot is taken and k is the rate constant at equilibrium for the pseudo-second order reaction. In the rate-controlling step, chemical reaction was found significant for all the factors studied. The R^2^ values at 0.025, 0.05, 0.075, and 0.1 g catalyst **SK9** doses calculated were around 1, as depicted in Figure 15. Meanwhile, R^2^ values for 50, 100, 150, and 200 ppm lignin concentrations were also around 1, as shown in Figure 16. Pseudo-second order kinetics offered a significant correlation between the experimental data and the reaction mechanism was evaluated to be chemically rate-controlled [58].

#### Activation Energy (E_a_)

During temperature treatment, the depolymerization stage is kinetically controlled by an energy barrier that encompasses multiple processes, like breaking of hydrogen bonds, water diffusion, and evaporation, among others [61]. Hence, activation energy values most probably depend upon the mechanistic clues of conversion, for example, the number of reaction steps involved in lignin depolymerization [62]. The pseudo-second order kinetic model was applied to data obtained during photo depolymerization of lignin with temperature variation and is depicted in Figure 17. It was observed that the depolymerization kinetic rate constant (k) correspondingly increased with increasing temperature, as given in Table 6.

By plotting the logarithm of apparent depolymerization kinetic constants (lnk) against 1/T, the linear curve was obtained, as depicted in Figure 18, with 0.97 R^2^ value following the Arrhenius equation (k = A^−Ea/RT^) rearrangement [63] as:lnk = −E_a_/RT + lnA(6)
where k is the reaction constant, E_a_ is the activation energy (kJ/mol), A is the frequency factor (h^−1^), R is the gas constant (8.314 J/mol·K) and T is the temperature (K). The E_a_ for lignin depolymerization calculated from the slope of the line (Slope = −E_a_/R) is 15 kJ/mol. Higher E_a_ indicates the depolymerization reaction was less influenced by temperature according to the rule. The calculated activation energy was remarkably lower in comparison to conventional lignin depolymerization, i.e., 59.75 kJ/mol [64]. Hence, a lower activation energy value signifies the efficiency of catalyst to depolymerize lignin more expediently.

### 2.4. Characterization of Depolymerization Products

#### GC-MS Analysis

The lignin depolymerization product (**L1-SK9**) was collected in liquid form after 60 min of photocatalytic activity of lignin (**L1**) with **SK9** under UV light at room temperature (20 °C). The second product **L1-SK9** (**90 °C**) was collected after 60 min of photocatalytic activity under the UV light of lignin by **SK9** at 90 °C to check the effect of higher temperature on lignin depolymerization. The GC is shown in Figure 19, which is characterized by the lignin solution peak before activity represented by ‘**A**’ in the chromatogram. Different peaks of lignin depolymerization products (**L1-SK9**) and **L1-SK9** (**90 °C**) represent the different monomeric lignin functionalities (listed in Table 7). MS corresponding to each GC signal is given in Appendix A with the structure of the expected compound present.

In the depolymerization product of lignin (**L1-SK9**), peak **1** (RT = 1.6) represents 18% butadienol (*m*/*z* = 71), which may be produced depolymerization of subunits between aromatic backbone of lignin [65,66]. Peak **2** (RT = 3.1) and Peak **3** (RT = 3.6) are attributed to 39% of 2-methoxy-4-methylphenol (*m*/*z* = 137) and 7% 2-methoxy-5-propenyl phenol (*m*/*z* = 164), which are interlinking subunits in the lignin structure [67]. Peak **4** (RT = 3.8) is due to 6% of 5-[2-(3-hydroxyphenyl)ethyl]-2-methoxyphenol (*m*/*z* = 244), which is basically the guaiacyl subunits of lignin structure [65]. On the other hand, in the depolymerization product of lignin **L1-SK9** (**90 °C**), peak **1′** (RT = 1.9) represents 14% methoxypropane (*m*/*z* = 75), which possibly originated due to further cracking of lignin interlinking subunits between the aromatic moieties due to temperature treatment. Peak **2′** (RT = 2.1) and peak **3′** (RT = 2.3) attributed to 17% of benzene (*m*/*z* = 78) and 19% cyclohexene (*m*/*z* = 81), whereas peak **4′** (RT = 2.6) and peak **5′** (RT = 2.8) are due to 10% phenol (*m*/*z* = 95) and 7% catechol (*m*/*z* = 110), which are part of aromatic backbone of lignin structure along with 16% 2-methoxy-4-methylphenol (*m*/*z* = 137 at RT = 3.1) represented by peak **6′**. From the GC-MS analysis, it was indicated that temperature treatment assisted in increasing catalytic depolymerization of lignin structural subunits into small monomeric functionalities, which was in accordance with %depolymerization obtained from temperature treatment during photocatalytic activity of lignin with **SK9**.

## 3. Materials and Methods

### 3.1. Chemicals and Reagents

Lignin was extracted from wood powder through the literature-reported organosolv acid treatment [68,69] reported in our recent study [70]. Lead chloride, HCl, and iso-propyl amine were purchased from Sigma Aldrich (Saint Louis, MO, USA). All of the chemicals were of analytical grade and utilized without further purification.

Powder X-ray Diffraction (PXRD) analysis was carried out in an X-ray diffractometer (Bruker, AXS D8; Yokohama-shi, Japan) with Cu-Kα radiation (1.54 Å) in the 2θ range from 20° to 60°. Scanning Electron Microscopy (SEM) (JEOL, JSM-6360 EO; Tokyo, Japan) analysis was performed to determine the morphological features of catalysts. Functional group characterization was done by an Infra-Red Spectrophotometer (Shimadzu; Osaka, Japan) in the range of wavenumber 500–4000 cm^−1^. Optical characterization was done by an Ultraviolet (UV) Spectrophotometer (Shimadzu; Osaka, Japan) in the frequency range of 250–800 nm and a Photoluminescence (PL) Spectrophotometer (Shimadzu; Osaka, Japan) at excitation wavelength 480 nm. Catalytic depolymerization was studied by taking absorption spectra by an Ultraviolet (UV) Spectrophotometer (Shimadzu; Osaka, Japan) in the range of wavelength 250–800 nm. Characterization of liquid lignin depolymerization products was done by Gas Chromatography Mass Spectrometry (GC-MS) (Shimadzu QP2010; Osaka, Japan, (MS Detector SPD 20A, Column: C-18 (250 × 4.6 mm)), Temperature; 300 °C). A UV Lamp (length: 288 mm, pipe diameter: 16 mm, voltage: 220 V, power: 8 W, wavelength range: 240–285 nm) was used as a light source for lignin depolymerization activity.

### 3.2. Synthesis of iso-Propyl Amine Lead Chloride Perovskites (IAPbCl_3_) (***SK9**–**SK10***)

**SK9** was prepared hydrothermally by taking 1 g PbCl_2_ salt into a Teflon lined autoclave container. Then, 50% 10 mL HCl and 4 mL iso-propyl amine (IA) were added. The container was tightly closed and heated at 150 °C for 24 h. The shiny white crystals of the obtained material were filtered and air-dried. The **SK9** obtained as white crystalline material which was characterized and used for catalytic activities. PXRD; D (77.19 ± 5). **SK10** was synthesized following the same way as for **SK9** by utilizing the 5 mL IA. The material **SK10** was also obtained as white crystalline material, which was characterized and used for catalytic activities. PXRD; D (75.59 ± 7).

### 3.3. Photocatalytic Activity under UV Light

Photocatalytic depolymerization of the lignin was done according to our recently reported photocatalytic activity under sunlight [70] modifying the light source in current studies. Lignin was extracted from the wood powder of *Oryza sativa* through the organosolv acid treatment method discussed in detail in our recent work about lignin depolymerization [70].

For photocatalytic activity experiments, 100 ppm lignin (**L1**) solution was prepared in dioxane. In total, 50 mL of this solution was stirred under UV light irradiation with 0.1 g amount of catalysts (**SK9** and **SK10**) separately. Aliquots of 5 mL from the reaction mixture were taken at time intervals of 10, 20, 30, 40, 50, and 60. A UV absorption spectrum was taken for all the taken aliquots at different time intervals. From the spectrum, the absorbance of samples at 280 nm was noted, and the %depolymerization for all the sample aliquots was calculated by following Equation (7) [26].
%Depolymerization = C_o_ − C_t_/C_o_ × 100(7)

In the equation, C_0_ is the starting concentration and C_t_ is the concentration of samples at a specific time interval. The lignin concentration at certain time intervals was calculated by drawing the calibration curve between the known lignin concentration and absorbance. The unknown concentration of lignin for noted absorbance at the different time intervals was measured by the linear equation of the known data.

The calculation of kinetic parameters was done by varying the initial lignin concentrations (50, 100, 150, and 200 ppm) and varying catalyst dosage (0.025, 0.05, 0.075 and 0.1 g) at the different time intervals through the same method as explained above. Meanwhile, thermodynamic parameters were evaluated by varying the temperature range (20, 35, 65, 90 °C) at different time intervals. Representative samples (**L1-SK9**, **L1-SK9** (temperature treated)) after the photocatalytic lignin depolymerization activity were analyzed by GC-MS.

## 4. Conclusions

The presented work described the synthesis of two iso-propylamine-based lead chloride perovskite nanomaterials (**SK9** and **SK10**) employed for photocatalytic applications. These perovskites were synthesized by employing a facile hydrothermal treatment and characterization was done by PXRD, SEM, UV, PL, and FTIR for determining the phase purity, structural composition, optical characteristics, and surface morphological features. Photocatalytic activity study was done by depolymerizing the lignin under UV light, which was extracted from the wood powder of *Oryza sativa*, reported in a previous work. Catalytic depolymerization of lignin by **SK9** and **SK10** was checked by absorption spectra at various time intervals and %lignin depolymerization was calculated. The effect of increasing catalyst dosage, starting lignin concentrations, and varying temperature indicated a considerable increase in %depolymerization of lignin. The kinetic study of investigated factors termed that pseudo-second order was the best suited kinetic model, with R^2^ > 0.9 representing that the reaction mechanism for lignin depolymerization was chemically controlled. The E_a_ for the reaction was calculated to be 15 kJ/mol, which is much less than conventional depolymerization reactions, i.e., 59.75 kJ/mol, and shows the good catalytic proficiencies of perovskites to depolymerize the lignin. Depolymerization products of the lignin were characterized by GC-MS analysis. GC-MS analysis of the liquid product of the lignin depolymerization was obtained with **SK9** at room temperature (20 °C) and at 90 °C, which indicated that temperature treatment assisted in increasing catalytic depolymerization of lignin structural subunits into small monomeric functionalities. Specifically, 39% 2-methoxy-4-methylphenol, 17% benzene, 10% phenol, and 7% catechol were confirmed by GC-MS analysis of lignin depolymerization.

## Figures and Tables

**Figure 1 molecules-25-03520-f001:**
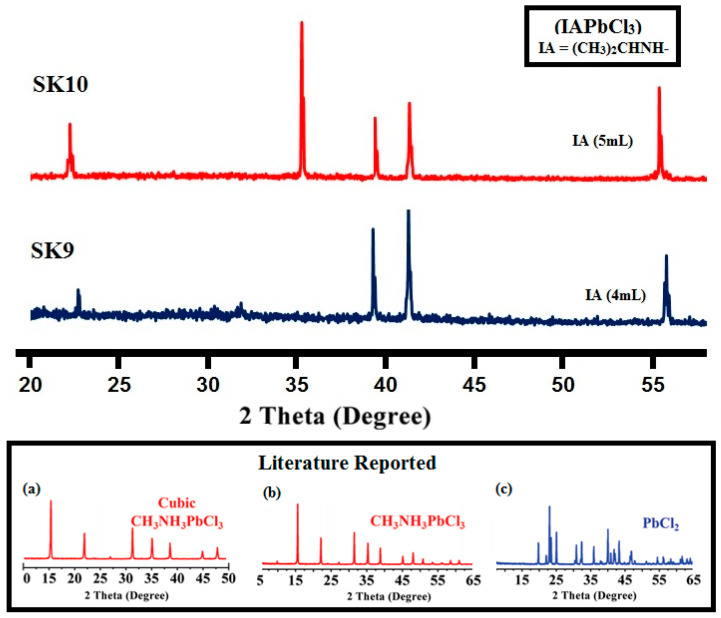
PXRD pattern of synthesized IAPbCl_3_ perovskite nanomaterials (**SK9**–**SK10**); Literature reported PXRD patterns of some methyl amine lead chloride perovskites are given in (**a**) and (**b**) and PbCl_2_ in (**c**) [36].

**Figure 2 molecules-25-03520-f002:**
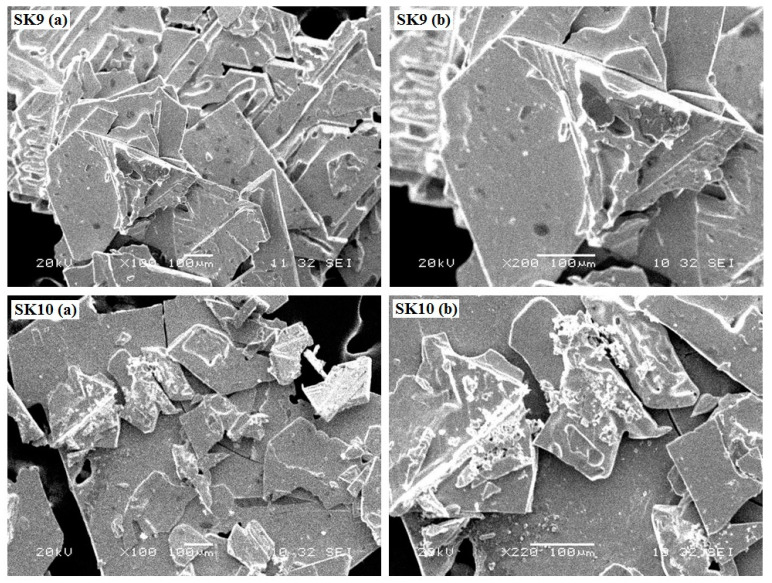
SEM images of synthesized perovskites **SK9-SK10**; [**SK9** (**a**): With ×100 Zooming; **SK9** (**b**): ×200 Zooming; **SK10** (**a**): With ×100 Zooming; **SK10** (**b**): ×220 Zooming].

**Figure 3 molecules-25-03520-f003:**
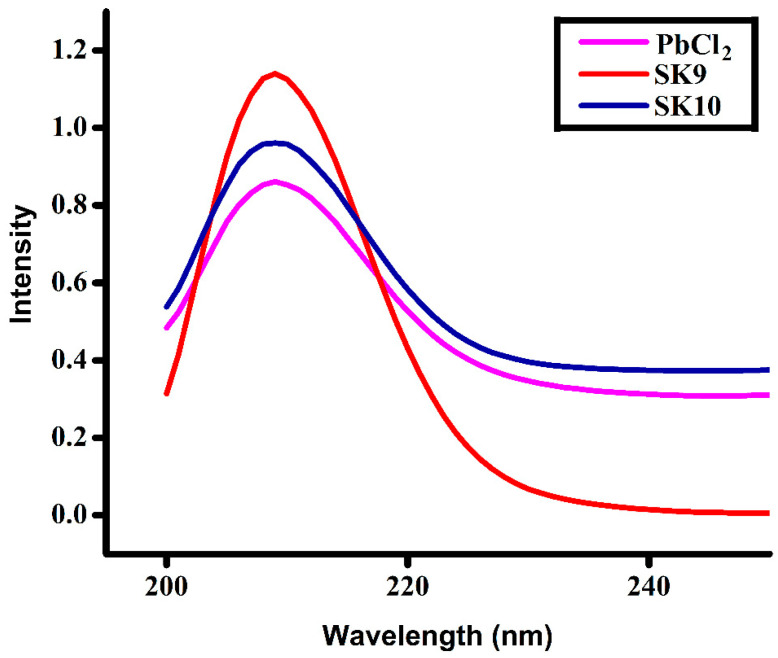
UV spectra of synthesized perovskites (**SK9**–**SK10**) and PbCl_2_.

**Figure 4 molecules-25-03520-f004:**
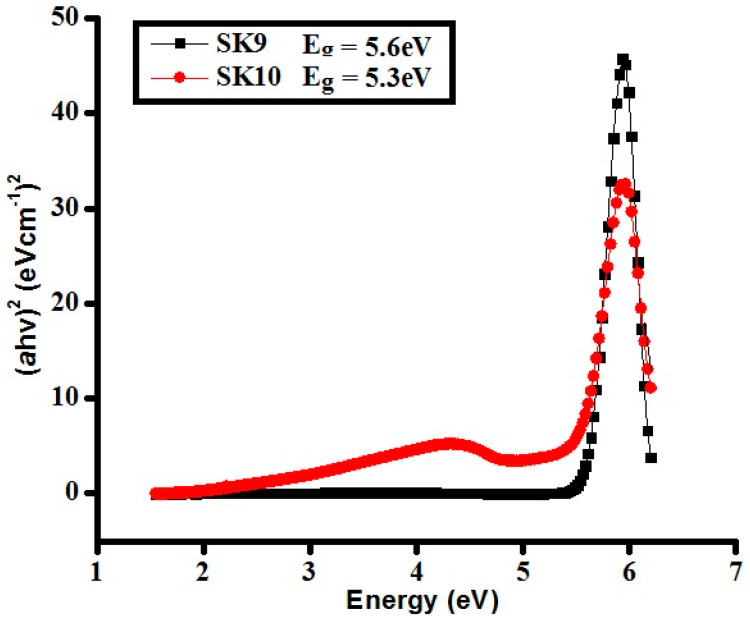
Tauc plots of synthesized perovskites (**SK9**–**SK10**).

**Figure 5 molecules-25-03520-f005:**
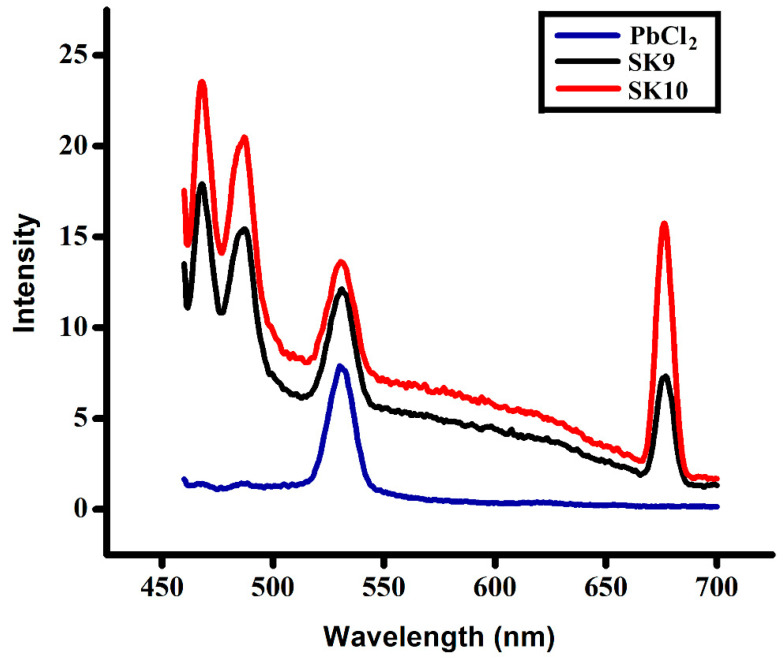
PL spectra of synthesized perovskites (**SK9-10**).

**Figure 6 molecules-25-03520-f006:**
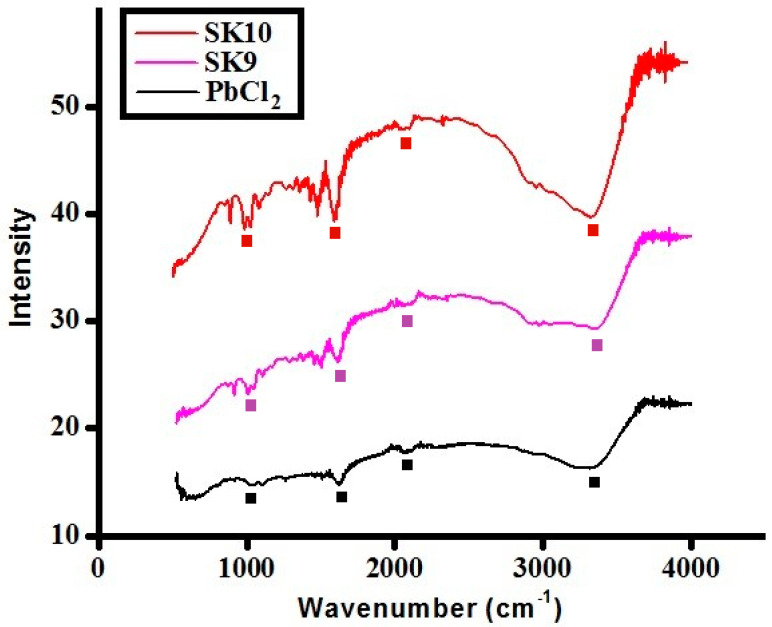
IR spectra of synthesized perovskites (**SK9**–**SK10**).

**Figure 7 molecules-25-03520-f007:**
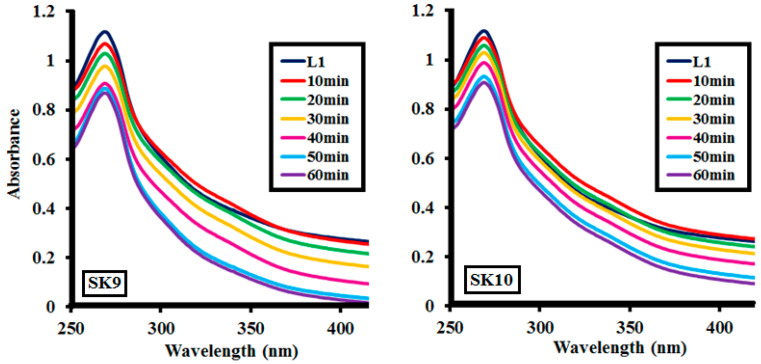
Photocatalytic lignin (**L1**_100ppm_) depolymerization by 0.1 g **SK9** and 0.1 g **SK10** at different time intervals.

**Figure 8 molecules-25-03520-f008:**
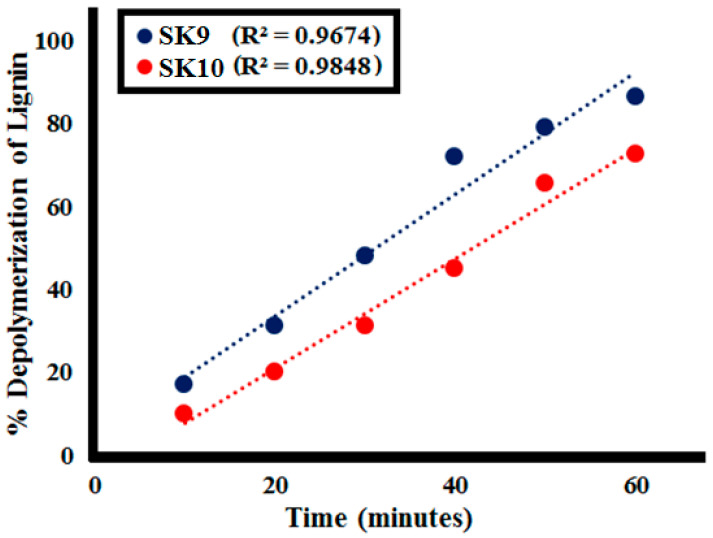
Correlation between the %depolymerization of lignin (**L1**_100ppm_) by **SK9** (0.1 g) and **SK10** (0.1 g) and the time intervals.

**Figure 9 molecules-25-03520-f009:**
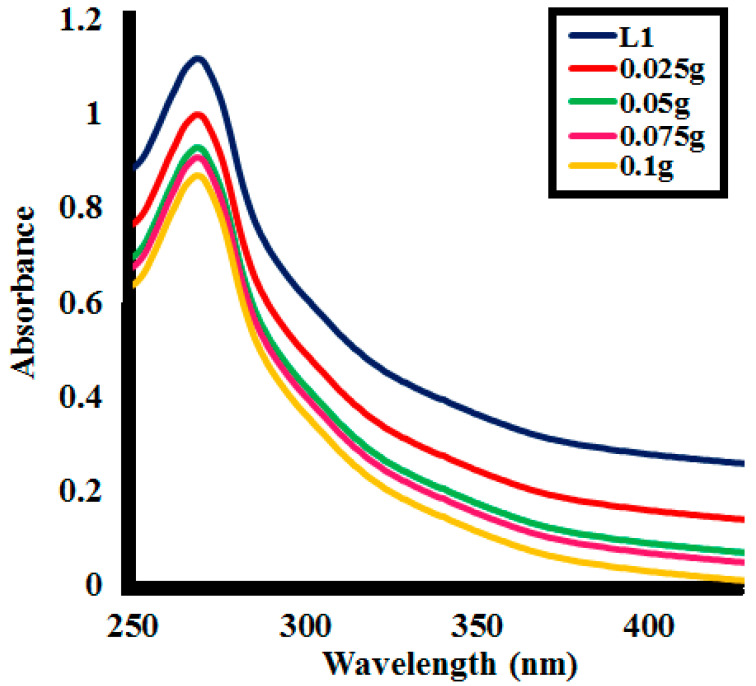
Photocatalytic lignin (**L1**_100ppm_) depolymerization by different catalyst doses of **SK9** after 60 min of catalytic activity.

**Figure 10 molecules-25-03520-f010:**
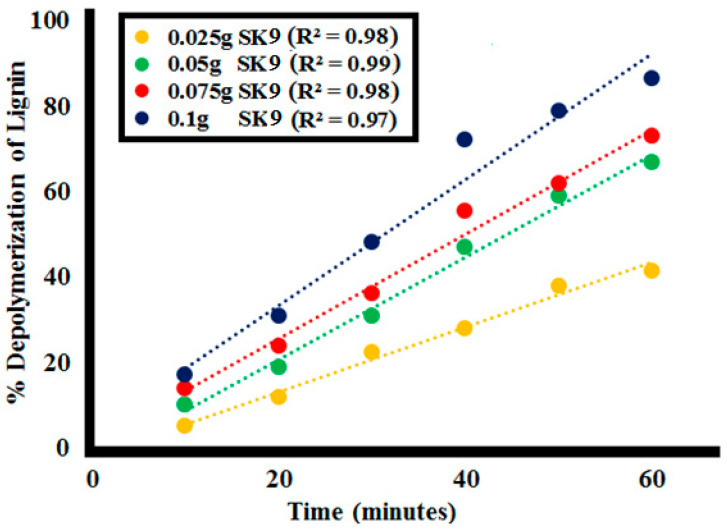
Correlation between %lignin (**L1**_100ppm_) depolymerization by **SK9** and the different catalyst doses at the different time intervals.

**Figure 11 molecules-25-03520-f011:**
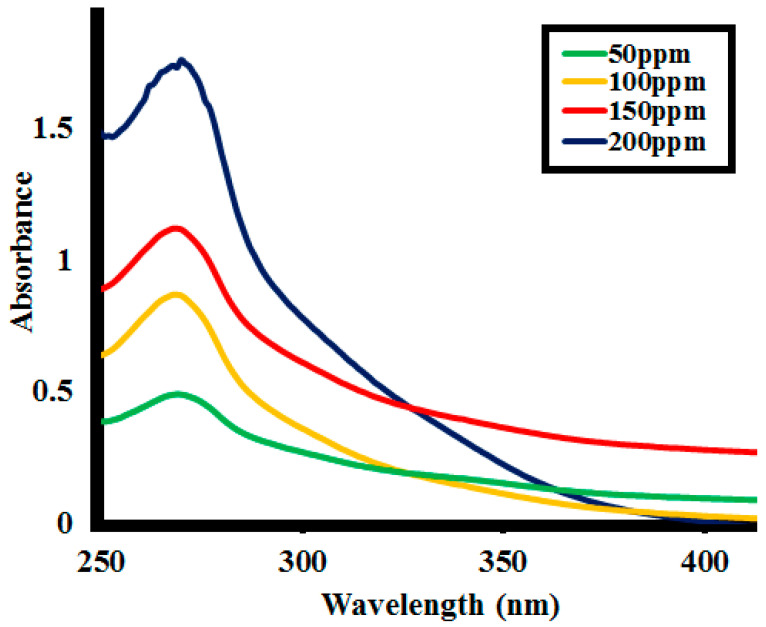
Photocatalytic depolymerization of the different lignin concentrations by 0.1 g **SK9** after 60 min of catalytic activity.

**Figure 12 molecules-25-03520-f012:**
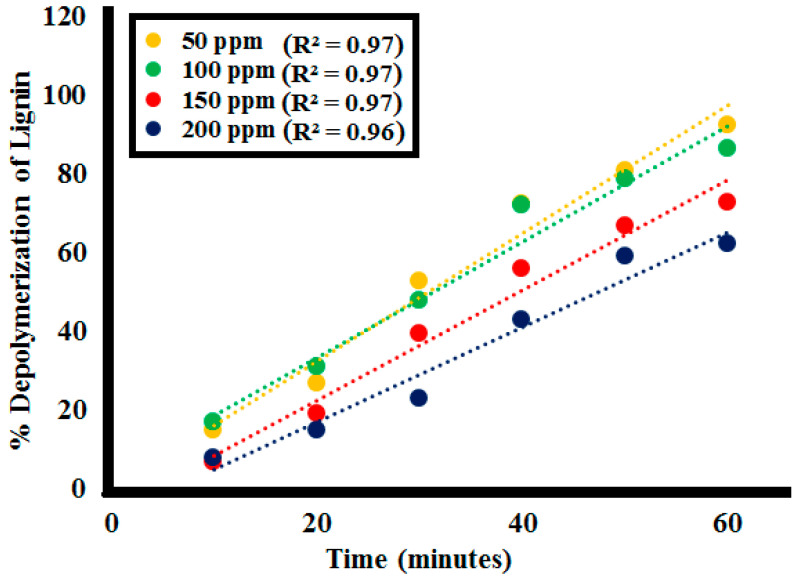
Correlation between %lignin (**L1**) depolymerization of the different lignin concentrations by **SK9** and the time intervals.

**Figure 13 molecules-25-03520-f013:**
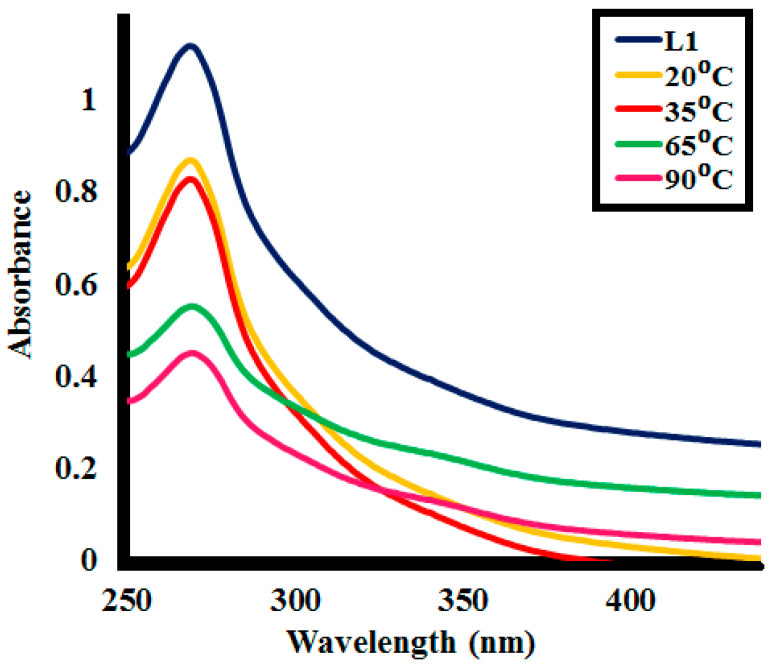
Photocatalytic lignin (**L1**_100ppm_) depolymerization by 0.1 g **SK9** at the different temperatures after 60 min of catalytic activity.

**Figure 14 molecules-25-03520-f014:**
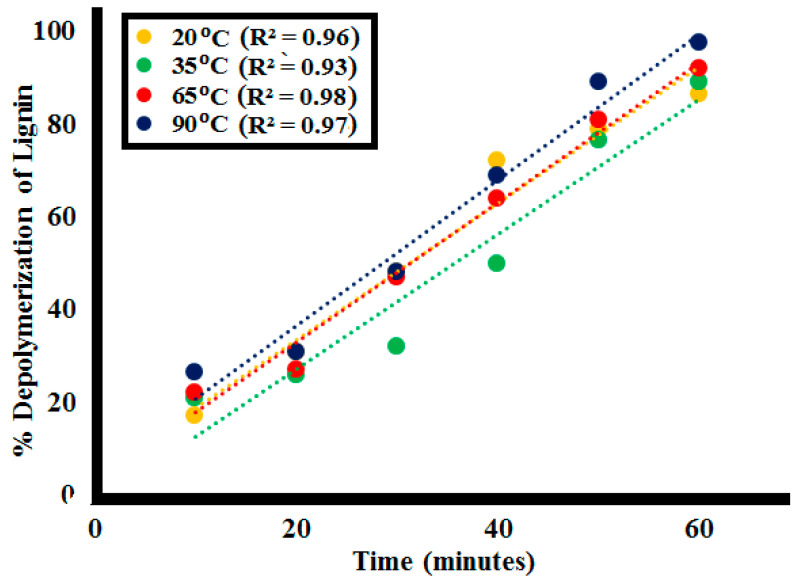
Correlation between %depolymerization of the lignin (**L1**_100ppm_) by **SK9** and different temperatures.

**Figure 15 molecules-25-03520-f015:**
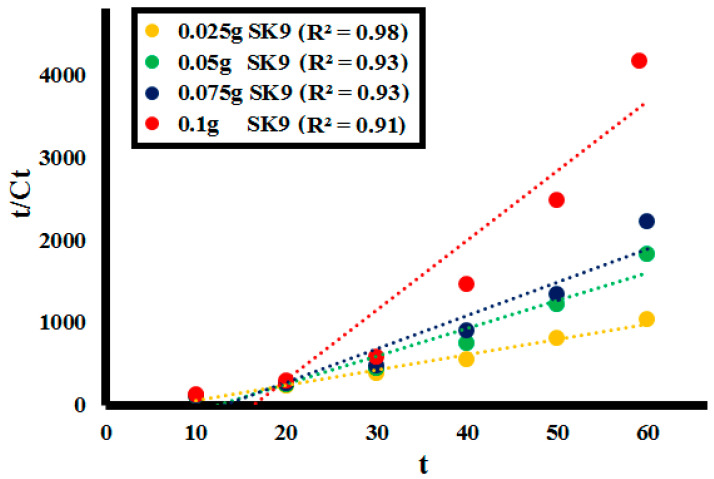
Pseudo-second order kinetics of the lignin (**L1**_100ppm_) depolymerization by different catalytic doses (0.025–0.1 g) of **SK9**.

**Figure 16 molecules-25-03520-f016:**
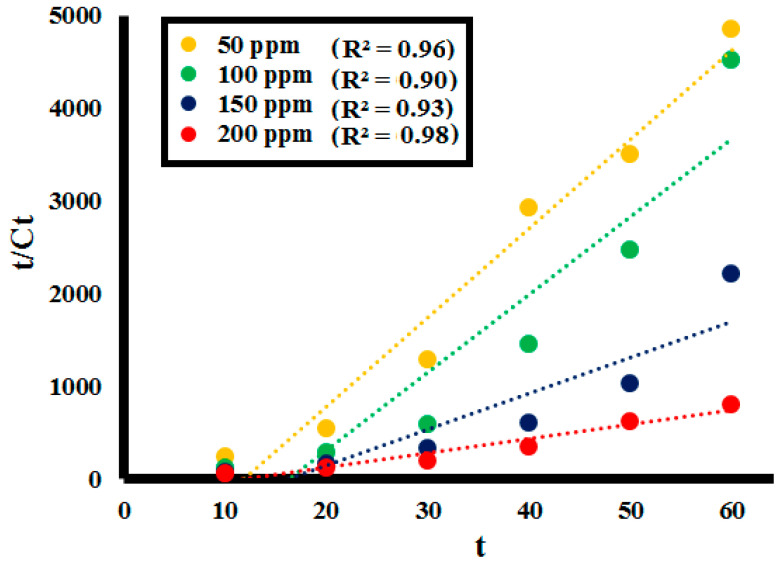
Pseudo-second order kinetics of the lignin (**L1**_100ppm_) depolymerization by different initial lignin concentrations (50–200 ppm).

**Figure 17 molecules-25-03520-f017:**
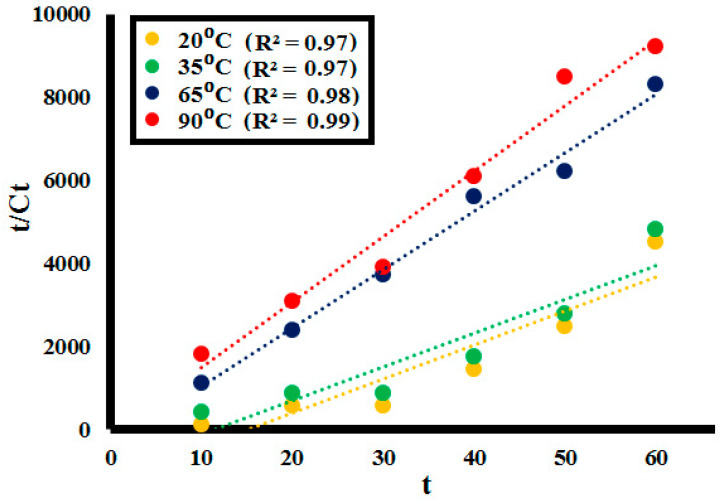
Pseudo-second order kinetics of the lignin depolymerization at different temperatures (20–90 °C).

**Figure 18 molecules-25-03520-f018:**
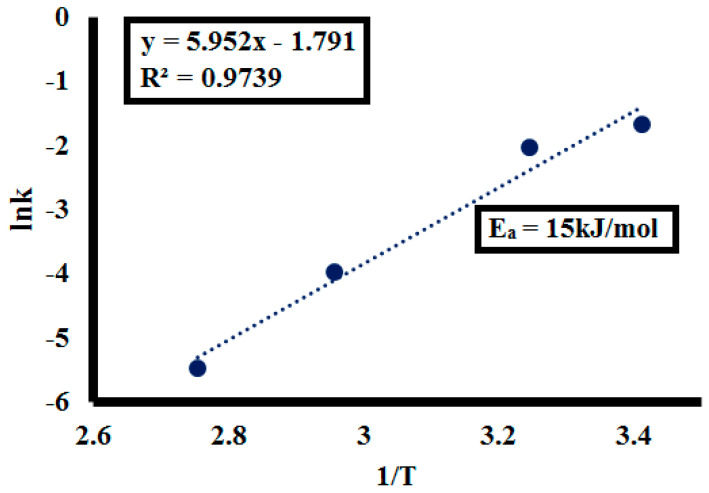
Arrhenius Law on temperature data.

**Figure 19 molecules-25-03520-f019:**
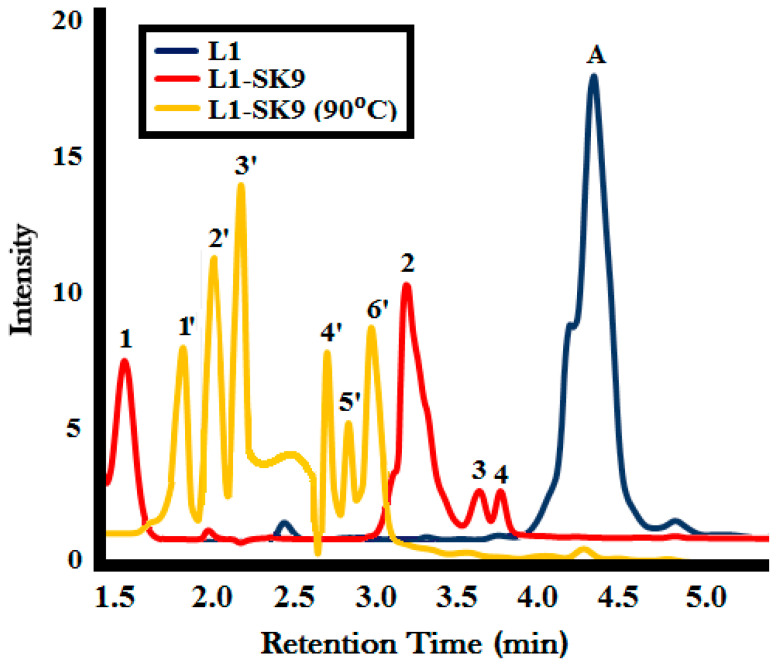
Gas spectrum (GC) of lignin (**L1**) and lignin depolymerization products **L1-SK9** and **L1-SK9** (**90 °C**); Samples after 60 min of photocatalytic activity under UV light at room temperature and at 90 °C, respectively.

**Table 1 molecules-25-03520-t001:** PXRD and SEM parameters of perovskite nanomaterials (**SK9** and **SK10**).

Samples	SK9	SK10
PXRD Parameters		
Average Crystallite Size D (nm)	77.10 ± 5	75.57 ± 7
Volume V = D^3^	1,162,604	1,051,104
Dislocation Density × 10^−3^ (nm)^−2^ (δ)	1.43 × 10^−7^	1.18 × 10^−7^
Microstrain (ε)	0.0017	0.0016
SEM Parameters		
Material Nature	Shiny Crystalline	Shiny Crystalline
Dispersity	Homogenous	Homogenous
Structural Appearance	Broad Elliptical Leaflets	Broad Elliptical Leaflets
Color	White	White

**Table 2 molecules-25-03520-t002:** %Lignin (**L1**_100ppm_) depolymerization by **SK9** (0.1 g) and **SK10** (0.1 g) at different time intervals.

Time	%Lignin (L1_100ppm_) Depolymerization at Different Lignin Concentrations	
(min)	SK9	SK10
0	0	0
10	17	10
20	31	20
30	48	31
40	72	45
50	79	65.5
60	86.5	72.5

**Table 3 molecules-25-03520-t003:** %Lignin (**L1**_100ppm_) depolymerization by different catalytic doses of **SK9** at different time intervals.

Time	%Lignin (L1_100ppm_) Depolymerization at Different Catalyst Doses of SK9			
(min)	0.025 g	0.05 g	0.075 g	0.1 g
0	0	0	0	0
10	5	10	14	17
20	12	19	24	31
30	22.5	31	36	48
40	28	47	55.5	72
50	38	59	62	79
60	41.5	67	73	86.5

**Table 4 molecules-25-03520-t004:** %Depolymerization of the different lignin concentrations by **SK9** (0.1 g) at the different time intervals.

Time	%Lignin (L1) Depolymerization at the Different Lignin Concentrations			
(min)	50 ppm	100 ppm	150 ppm	200 ppm
0	0	0	0	0
10	15	17	7	8
20	27	31	19	15
30	53	48	39.5	23
40	72.5	72	56	43
50	81	79	67	59
60	92.5	86.5	73	62.5

**Table 5 molecules-25-03520-t005:** %Lignin (**L1**_100ppm_) depolymerization by **SK9** (0.1 g) at the different temperatures and the time intervals.

Time	%Lignin (L1_100ppm_) Depolymerization at Different Temperatures			
(min)	20 °C	35 °C	65 °C	90 °C
0	0	0	0	0
10	17	21	22	26.5
20	31	26	27	31
30	48	32	47	48
40	72	50	64	69
50	79	76.5	81	89
60	86.5	89	92	97.5

**Table 6 molecules-25-03520-t006:** Pseudo-second order kinetic parameters at different temperatures.

Pseudo Second Order Kinetic Parameters at Different Temperatures		
Temperature	Rate Constant (k) (g^−1^min^−1^)	R^2^
20 °C	0.011	0.97
35 °C	0.018	0.97
65 °C	0.132	0.98
90 °C	0.186	0.99

**Table 7 molecules-25-03520-t007:** GC-MS data of the lignin depolymerization products **L1-SK9** and **L1-SK9** (**90 °C**).

GC Peak No.	Retention Time (RT)	*m*/*z* of Molecular Ion Peak	Name	Concentration (%)
**L1-SK9**				
**1**	1.6	71	Butadienol	18
**2**	3.1	137	2-methoxy-4-methylphenol	39
**3**	3.6	164	2-methoxy-5-propenyl phenol	7
**4**	3.8	244	5-[2-(3-hydroxyphenyl)ethyl]-2-methoxyphenol (guaiacyl dimer)	6
**L1-SK9** (**90 °C**)				
**1′**	1.9	75	Methoxypropane	14
**2′**	2.1	78	Benzene	17
**3′**	2.3	81	Cyclohexene	19
**4′**	2.6	95	Phenol	10
**5′**	2.8	110	Catechol	7
**6′**	3.1	137	2-methoxy-4-methylphenol	16

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
