# Peer review of "i-Propylammonium Lead Chloride Based Perovskite Photocatalysts for Depolymerization of Lignin Under UV Light"

_molecules, 2020, doi:10.3390/molecules25153520_

Round 1

Reviewer 1 Report

The manuscript describes the i-propylammonium lead chloride based Perovskite photocatalysts for depolymerization of lignin under UV light. The work is interesting.

The paper contains elements of novelty and may be published but after major revisions. I would recommend the following amendments:

-Please specify what lignin was used (from which trees) because I understand that the authors isolated the lignin themselves? Please enter some procedure (shortly).

- Full lignin characteristics should be added, among others molecular weight and amount of hydroxyl and phenol groups. This is very important because the authors state that received as much as 18% butadienol.

-Lignin strongly absorbs UV radiation, have the authors observed this phenomenon? If so, does it not negatively effect on the depolymerization process ?

-Figure 6 should be corrected, the most important signals should be marked on the curves.

-Please indicate what wavelength of light was used, as well as the light source and size of the lignin vessel. Has lignin dissolved in dioxide?

Reviewer 2 Report

In this study, photocatalytic depolymerization of lignin has been investigated by two new iso-propylamine based lead chloride perovskite nanomaterials (SK9 and SK10) synthesized by facile hydrothermal method. Some comments are given below:

1) In Abstarct the abbreviated words must be given in full details also (eg. Scanning Electron Microscopy (SEM))

2) The novelty of this work must be further emphasized in Introduction.

3) The discussion of characterizations must be improved.

Reviewer 3 Report

Submitted manuscript describes new isopropylamine lead chloride materials used as photocatalysts for lignin depolymerization. The subject is topic and I recommend its publication after minor revision.

  1. Eq.(4) is somehow damaged, please check it.
  2. Similarly Eq. (5) needs to be checked and equipped with explanation of variables.
  3. Table 6 – there are probably rate constants not reaction rates. Please add units to the constants k.
  4. Ea for lignin depolymerization was calculated as 11kJ/mol (p.19, line 5 from bottom), however, elsewhere the value 15 kJ/mol is given. What is true?
  5. The characterization of UV light irradiation during photocatalytic experiments should be given – the source of lights, wavelengths range, intensity.
  6. The products of lignin depolymerization were analyzed by GC-MS. Please give conditions (column, temperature program etc.) and explain how quantification was achieved.

Round 2

Reviewer 1 Report

Accept in present form.

Reviewer 2 Report

All my comments of the initial submission have been correctly replied and included in the revised manuscript. The quality of this work has been drastically improved after revision and therefore I recommend its publication as it is.